# Current and Future Flow Cytometry Applications Contributing to Antimicrobial Resistance Control

**DOI:** 10.3390/microorganisms11051300

**Published:** 2023-05-16

**Authors:** Luminita Gabriela Marutescu

**Affiliations:** 1Department of Botany and Microbiology, Faculty of Biology, University of Bucharest, 91-95 Spl. Independentei, 050095 Bucharest, Romania; luminita.marutescu@bio.unibuc.ro; 2Research Institute of the University of Bucharest, 050095 Bucharest, Romania

**Keywords:** one health, AMR challenge, flow cytometry, clinical samples, environment

## Abstract

Antimicrobial resistance is a global threat to human health and welfare, food safety, and environmental health. The rapid detection and quantification of antimicrobial resistance are important for both infectious disease control and public health threat assessment. Technologies such as flow cytometry can provide clinicians with the early information, they need for appropriate antibiotic treatment. At the same time, cytometry platforms facilitate the measurement of antibiotic-resistant bacteria in environments impacted by human activities, enabling assessment of their impact on watersheds and soils. This review focuses on the latest applications of flow cytometry for the detection of pathogens and antibiotic-resistant bacteria in both clinical and environmental samples. Novel antimicrobial susceptibility testing frameworks embedding flow cytometry assays can contribute to the implementation of global antimicrobial resistance surveillance systems that are needed for science-based decisions and actions.

## 1. Introduction

Flow cytometry (FCM) is a powerful, quantitative, single-cell technology that has enabled important breakthroughs in various scientific fields such as immunology, cell biology, oncology, medical microbiology, environmental microbiology, and the food industry since its development in the 1960s. The ability to analyze microbial cells without culturing, with high accuracy, rapid analysis, and multi-parameter measurements, as technical innovations have significantly advanced the field of cytometry in life sciences and biomedicine. Recently, the antimicrobial resistance (AMR) challenge has led scientists to advance new proof-of-concept approaches, as well as commercial testing based on FCM assays for rapid and direct detection of antibiotic-resistant bacteria in clinical samples, detection of antibiotic-resistant bacteria in environmental samples (soil, water, air), and research on the spread and persistence of resistance [1,2] in environments influenced by human activities. This review focuses on the latest FCM applications, both alone and in combination with other technologies, to address the AR challenge. Future prospects for FCM applications are also discussed, including embedding cytometry in protocols for integrated surveillance of AMR, improving current antimicrobial therapies for polymicrobial infections and biofilm-associated infections, and the potential role of FCM in understanding the spread and evolution of AMR in the environment.

## 2. The AR Challenge

Despite all the global collaborative efforts that began with the WHO Global Action Plan on AMR in the year 2015 and continuing today with the Tripartite Agreement (WHO, FAO, and OIE), rates of AMR continue to rise. A comprehensive analysis of the global burden of AMR has shown that it is the leading cause of death worldwide. In 2019, a total of 3.57 million deaths were associated with AMR, and it is estimated that by 2050, 10 million people will die from infections produced by drug-resistant pathogens each year. This would even surpass cancer as the leading cause of death worldwide [3]. AMR will also impact routine medical procedures such as organ transplantation, childbirth, and chemotherapy, and increase healthcare costs [4]. To address and overcome this urgent global challenge of our time, greater collaboration among governments, agencies, civil society, and the private sector is needed to ensure sustainable and effective global action.

At the global level, antibiotic use increased between 2000 and 2015, and this growth is expected to continue [5]. The massive and widespread use of antimicrobials in clinical settings [6,7], municipal settings [8], livestock, crop production, and aquaculture is considered one of the main drivers of the increase in this otherwise natural response of microorganisms [9]. However, over the past seven years, analysis of the AMR challenge has contributed to a better understanding of its broader dimensions. Social and economic determinants have been identified as important enablers and facilitators of AMR occurrence and spread [10,11]. The multifaceted nature of AMR requires efforts to permanently integrate it into national and international sustainability and development agendas [12].

Given the global AMR challenge, several mitigation strategies, summarized in the “One Health” concept, could help reduce the AMR burden. These strategies include the reduction of antibiotic use in nonmedical settings (agriculture, farm animals, and aquaculture) and rational use of antibiotics in healthcare settings and the community by appropriate prescribing. Additionally, there is an urgent need for prioritizing research and development strategies focused on novel antibiotics, particularly against multidrug-resistant tuberculosis and Gram-negative bacteria [13], and also on alternatives to antibiotics. At the same time, knowledge of the potential mechanisms of AMR and its spread in microorganisms can support the development of mitigation measures to protect the efficacy of antimicrobials.

## 3. The Landscape of Rapid Methods for Antibiotic Susceptibility Testing

There is clearly an urgent need for the further development of innovative approaches to obtain faster results on antibiotic susceptibility of pathogens [14,15,16,17,18]. Rapid antibiotic susceptibility testing (AST) will help physicians rapidly treat patients with the right antibiotic and limit the spread of antibiotic resistance. Currently available standard ASTs, based on broth or agar dilution, require isolation of bacteria in pure culture followed by identification and quantitative assessment of growth inhibition against a panel of antibiotics. Their major limitation is that results for most clinically important bacteria are available within 18–24 or 72 h.

Critically ill patients with severe infections such as bloodstream infections and sepsis require rapid and effective antibiotic therapy. Until recently, these therapies were based on predictable antibiotic profiles. However, the increase in AMR rates of the pathogens and the delay of microbiological results make empirical antibiotic therapy challenging [18,19]. Notably, it has been reported that one in five patients with bloodstream infections in US hospitals received inappropriate antibiotic treatment [19]. The development of rapid tests for pathogen identification and detection of AMR will enable the determination of which antimicrobials should be used for treatment. Preliminary results on the pathogen’s antibiotic profile will contribute to the reduction of antibiotic use and will help achieve multiple clinical goals, including improving treatment outcomes [20] (speed and accuracy), reducing exposure to toxic antibiotic regimens, limiting the selection of AMR pathogens, and reducing healthcare costs.

Based on their turnaround time, there are rapid diagnostic methods AST that allow us to identify high-risk pathogens and determine whether they are drug susceptible or drug resistant in less than 8 h. Ultrafast (direct) AST or point-of-care antibiotic resistance diagnostics allow direct analysis of polymicrobial samples without pre-cleaning steps and provide results in less than 4 h. However, to date, no method exists that allows for short (approximately 15 min) antibiotic exposure and short (approximately 15 min) test duration without complex instrumentation, allowing the method to be used at the point of care.

In general, rapid methods AST can be divided into either genotypic or phenotypic methods. Genotypic AST methods detect resistance by screening specific genetic resistance markers. They can provide faster results because, unlike traditional AST methods, they do not rely on microbial growth and can also be performed directly on biological samples and hemocultures, reducing turnaround time. However, there are hundreds of genes that confer resistance, and current rapid genotypic tests detect only some of them. Moreover, these tests do not provide information on the phenotypic expression of resistance genes. Several excellent detailed reviews of current and emerging rapid genotypic tests have been published [21,22,23,24,25,26,27]. However, very few molecular ASTs have received FDA approval.

Phenotypic ASTs detect AMR by measuring the pathogen’s response to antibiotics. The ASTs that are the gold standard, including the disc diffusion test, E-test, agar and broth dilution methods, are phenotypic tests that measure microbial growth in the presence of antibiotics. However, they take several days to provide a result because they require the isolation of bacteria isolated from patients who were then exposed to the antibiotics. To overcome this delay in conventional AST assays, new phenotypic strategies have been proposed for the rapid measurement of various phenotypic characteristics of bacteria, such as morphology, metabolism, biochemical composition, and growth after exposure to antibiotics, including isothermal microcalorimetry [28,29], electrochemical impedance spectroscopy [30,31], microscopy [32,33], electrochemical ASTs [34,35], spectroscopy [36], flow cytometry [37,38,39,40,41,42,43,44,45], and spectrometry [46]. Rapid phenotypic AST methods have detected microbial growth and/or metabolism as well as morphological changes in biological samples with very low microbial loads. Thus, the most important prerequisite for rapid phenotypic AST methods is that they must have sufficient measurement sensitivity. To meet this challenge, various innovations in electronics, biosensors, optics and microfluidics are required.

## 4. Rapid Detection of Bacterial Pathogens and Resistance–Contribution of FCM

FCM evaluates numerous physical and chemical properties of individual cells or particles flowing in a fluid stream. A series of detectors are used to evaluate scattering and fluorescence (when cells are fluorescently labeled), and the resulting analysis is used to create multi-parameter data sets that depict the physical characteristics of the cells and their fluorescence properties. The size and complexity of the cells are evaluated based on their forward and side light scatter measurements. Staining with fluorescently tagged antibodies or dyes that detect cellular components and/or integrity (viability) aids in the detection and quantification of cellular characteristics and/or expression of various proteins. One key advantage of this technique is the ability to perform these measurements in a very rapid time span. In a single sample, up to twenty cell proprieties can be assessed, individually, cell by cell, for 10,000 cells, in less than a minute.

Several studies have shown that FCM assays are a promising tool in clinical microbiology, as they can be used to determine whether the bacterium is sensitive, resistant, or intermediate to a particular antibiotic [40,41]. Table 1 provides examples of FCM-based AST approaches developed for the detection of AMR in pathogenic bacteria. FCM assays were demonstrated to accurately and rapidly detect AMR profiles of pathogens directly in clinical samples [37,38,42,47,48,49,50,51,52,53]. Most of the developed AST methods based on FCM used fluorescent dyes to assess the viability of microbial cells after exposure to antibiotics. However, the progress of phenotypic ASTs based on FCM assays has been hindered by the different interactions between bacteria and antibiotics, the nonspecific binding of fluorescent dyes, and the limited computational power to detect changes within heterogeneous populations [18]. However, recently, commercial ASTs based on FCM have been developed to provide AST reports within 2 h, instead of 24–48 h in the case of current standard methods [39,40,41,42].

ASTs based on FCM methods offer the great advantage of a short indication time for the microbial mortality level in clinical samples exposed to antibiotics, in contrast to classical culture-based methods that take a minimum of 24 h (Table 2). FCM was shown to be able to detect bacterial pathogens in clinical urine samples and to differentiate among their susceptibility or resistance to ciprofloxacin, trimethoprim–sulfamethoxazole nitrofurantoin, and ceftriaxone after 4 h of incubation versus the minimum of 24 h required by the conventional ASTs test based on cultivation [37]. Molecular ASTs are faster than plating and can predict phenotypic drug resistance but with variable sensitivity. Additionally, these assays are not able to detect emerging new resistance mechanisms; therefore, phenotypic testing remains essential.

Most FCM assays evaluate microbial cell viability based on analysis of various structural and/or functional characteristics such as membrane integrity, metabolic activity, and membrane potential using non-specific fluorescent dyes, including nucleic acid dyes such as propidium iodide, YO-PRO-1, acridine orange, thiazole orange, and SYTO-9, membrane potential sensitive dyes such as DiBAC4(3), and esterase activity such as fluorescein diacetate [58]. By combining fluorescent dyes, FCM assays allow for the detection of intermediate physiological states between viable and dead cells, highlighting the inherent heterogeneities of microbial populations. Based on functional and structural characteristics, Nebe-von Caron et al. (2000) [59] identified viable, metabolically active, intact, and permeabilized cells. This information cannot be obtained by standard plate counts or biochemistry. Figure 1 depicts the heterogeneous response of *E. coli* reference strain against a commercial biocidal product (at MIC concentration), used for surface disinfection in healthcare settings.

### 4.1. Direct Pathogen Detection and AST on Clinical Samples Using FCM

#### 4.1.1. Urine Samples

Diagnosis of urinary tract infections, the most common infectious disease in both community and hospital settings, affecting 150 million people worldwide each year [60], has been improved by the development of automated FCM analyzers [61]. FCM analyzers improve the speed and accuracy of urinalysis by determining Gram-negative bacteria with high sensitivity and high agreement in comparison with cultivation methods [62,63]. Based on scatter analysis, it was shown that FCM could classify bacterial groups in fresh positive urine samples collected from outpatients into the cocci and bacilli or polymicrobial groups, but it could not differentiate them by strain. The proposed workflow, which incorporated FCM and dip-stick test, could reduce the use of antibiotics such as fluoroquinolones and limit the emergence of drug-resistant bacteria [63].

Studies reported the use of FCM for the selection of positive urine specimens, followed by bacterial identification with MALDI-TOF MS and then direct testing AST with the VITEK 2 system to reduce the waiting time for results by up to one day [54,64]. In addition, FCM combined with DiBAC4(3), a membrane potential-sensitive dye, enabled direct AST in urine samples [37]. After 4 h of incubation with antibiotics (ciprofloxacin, ceftriaxone, nitrofurantoin, trimethoprim–sulfamethoxazole), the urine samples were directly analyzed using FCM. Measurement green fluorescence distribution of DiBAC4(3) that accumulates inside the cell upon membrane potential decrease, allowing for the differentiation, directly in urine samples, between antibiotic-sensitive and resistant uropathogenic *E. coli* isolates [37]. By providing preliminary information on AMR patterns of uropathogens, rapid direct AST approaches using FCM contribute to improving early empirical therapy of urinary tract infections and preventing the emergence of AMR bacteria.

#### 4.1.2. Blood Samples and Hemocultures

Bloodstream infections caused by multidrug-resistant bacteria are associated with many deaths worldwide [65]. Rapid direct detection AST of blood samples will support appropriate therapy, reduce drug misuse, and, more importantly, improve clinical outcomes [41]. However, unlike urine samples, the ~10^9^ blood cells/ml interferes with the typical low bacterial load of ≤100–1000 bacterial cells/ml blood in septic patients. Nevertheless, bacteria have been detected directly in blood samples using FCM [38,66], microfluidics [67,68,69], and PCR [70,71]. Huang et al. (2018) [38] developed a free-label FCM combined with an adaptive multidimensional statistical metric called probability binning signature quadratic form (PB-sQF) for direct AST in blood samples, bypassing the tedious blood culture-based amplification. The proposed procedure involved lysis of blood cells, recovery of bacteria, pre-incubation for 2 h and another 3 h for bacterial ‘exposure to antibiotics’ followed by FCM analysis. Blood samples spiked with *E. coli*, *Klebsiella pneumoniae*, or *Acinetobacter nosocomialis* were used for the procedure, and resistance profiles to various antibiotics were determined based on scatter data analysis. The time to result was reduced from >60 h to <8 h. In another study, Gu et al. (2019) [55] used MALDI-TOF MS and FCM for rapid identification and AST for positive blood cultures. The research results showed that the FCM AST results for the 238 pathogenic bacteria were consistent with the VITEK2 system results, demonstrating the reproducibility of FCM compared to standard automated systems for AST. The reporting time could be reduced to 3 h. There are several limitations to the approaches developed; however, the preliminary rapid results could help reduce antibiotic misuse and thus the emergence and spread of AMR bacteria.

To help physicians provide appropriate antibiotic treatment in a timely manner, FASTinov, a spin-off from the University of Porto (Portugal), has recently developed commercial FCM kits for AST testing of positive blood cultures. Metabolic changes following exposure to a range of antibiotics are detected in less than two hours using FCM in conjunction with fluorescent dyes [39,40,41]. To overcome the nonspecific dye-bacteria interactions, Filbrun et al. (2022) [53] developed a label-free FCM approach for the rapid assessment of bacterial antibiotic susceptibilities directly from positive blood culture. The FCM method was used to determine antibiotic-induced changes in count rate, taking into account the scatter position. The results showed that 90% of antibiotic susceptibility could be determined within only 5 h after positive blood culture.

#### 4.1.3. Peritoneal Dialysis and Sputum

Peritonitis is a common and serious complication of peritoneal dialysis. It can become rapidly life-threatening without effective antimicrobial therapy. To improve diagnosis and patient outcomes, there is an urgent need for preliminary data on bacterial pathogens to guide antimicrobial prescription practices in an appropriate time frame. A case report showed that the FCM technique was able to detect and quantify bacterial load within 2 h of receiving the clinical specimen [72]. Moreover, the FCM method has demonstrated a strong positive correlation with culture-dependent methods, thus being useful for common peritoneal dialysis pathogens across a range of relevant antimicrobials [73].

FCM has significantly improved our understanding of the physiology and pathology of *M. tuberculosis* in response to drug therapy [74,75]. Fluorogenic probes, such as 4-N, N-dimethylamino-1,8-naphthalimide [76] or 3-hydroxychromone (3HC) [77] in combination with microscopy or flow cytometry, enabled the rapid detection of live and dead *M. tuberculosis* cells in TB patient sputum samples. Rapid and accurate detection of mycobacteria directly in sputum samples could improve the efficacy of antibiotic treatment. Moreover, FCM-based phenotyping and whole-genome sequencing were shown to detect *M. tuberculosis* drug resistance [75].

Overall, FCM technology shows potential for broad applicability in emergency care settings [42]. The use of FCM-based testing to detect infections and preliminary AST results can support rapid treatment with the right antibiotic, limiting the emergence of AR and improving patient outcomes [39,40,41,42] (Figure 2). Innovative, rapid, and efficient diagnostic tools, such as FCM, could support personalized antibiotic recommendations, offering a means to reduce the emergence and spread of resistant pathogens (Figure 3).

### 4.2. Detection and Quantification of AMR in the Environment

Overuse and misuse of antibiotics in healthcare settings and commercially driven agriculture (including livestock and fish farming) led to an increase in antibiotic rates in pathogens and their spread in the environment [4]. Our water sources are exposed to antibiotic residues, antibiotic-resistant bacteria and antibiotic resistance genes, and other pharmaceuticals which are discharged into the receiving environment through effluents of wastewater treatment plants (WWTP) that are not completely removing these pollutants during treatment [78]. Quantification of AMR in water is required for an efficient risk assessment in environmental reservoirs [79]. Liang et al. (2020) [80] developed integrating metagenomics and FCM analysis applicable to aquatic microbiological risk assessment. The approach allowed for the identification and quantification of pathogenic bacteria carrying both AMR genes and virulence factor genes in the environment, which is of particular concern due to their infection ability and AMR. Compared with culture-based methods, the developed approach provides fast results, more feasible for large-scale environmental surveys. The shortfall of other molecular methods, such as qPCR and amplicon next-generation sequencing is that the genomic context of the target gene is missing. Thus, the methodology that integrated FCM could be useful for the identification and enumeration of antibiotic-resistant pathogens in the environment.

Strong evidence supports that activated sludge in WWTPs acts as the hotspot for harboring and amplifying AMR [81]. Recently, Miłobedzka et al. (2022) [82] published an extensive review that provides an overview of distinct approaches that are used or can be adapted to monitor antibiotic resistance genes in wastewater environments. The authors also highlighted that the assessment of the diversity and abundance of antibiotic resistance genes is insufficient to provide information with regard to the direct or indirect risks for human health. Single-cell analysis approaches that by-pass cultivation will help identify the links between AMR genes and MGEs and the range of specific microbial hosts. Qiu et al. (2018) [83] reported an agarose-based microfluidic chip integrated with FCM assay and high-throughput sequencing for determining and quantifying the transfer rate of plasmid-carrying antibiotic resistance genes in sludge biofilms. Such information is critical for controlling AMR. Understanding the mechanisms of acquisition and transmission of AMR is critical to stop the increasing AMR threat.

Manuring was shown to increase the abundance of AMR bacteria in soil [84,85,86]. The assessment of the risks of AMR spread via manure application on land is essential for controlling AMR. FCM assays were used for assessing the AMR gene transfer from fecal to soil bacteria [87]. The researchers used the FCM approach and plate count for the enumeration of transconjugants. The advantage of FCM assay in comparison with the cultivation method is that it can rapidly detect viable non-culturable cells. However, the authors did not find differences between plate counts and FCM results with regard to the abundance of transconjugants, indicating that most of them were culturable [87]. Recently, the FCM approach, coupled with transmission electron microscopy, RT-qPCR, and RNA-seq techniques, was used to assess the impact of fungicide exposure on AMR dissemination. Zhang et al. (2023) [2] revealed that the frequency of conjugative transfer was correlated with increased exposure concentrations of chlorothalonil, azoxystrobin, and carbendazim, suggesting the potential role of pesticides on the AMR spread.

It is clear that there is a link between AMR bacteria in the environment and anthropogenic contamination [88,89,90,91,92,93]. Recently, FCM coupled with qPCR was used for the enumeration of total airborne bacteria and quantification of AMR genes in marine and continental clouds [88]. The study provided evidence that clouds represent a potential reservoir of ARGs within the environment [94]. Scott et al. (2021) [95] used FCM for the detection and quantification of AMR bacteria in soil and water samples collected from Rocky Mountain National Park in the United States. Using high-throughput FCM, the researchers revealed the presence of AMR bacteria in all environmental samples tested. These results suggest that human presence drives the abundance of AMR bacteria in the natural environment. FCM analysis platforms could be used to measure AMR bacteria as indicators of human activity and also to evaluate their impact on watersheds and soils.

Altogether, these reports are suggesting that FCM assays can support scientists in monitoring the AMR gene transfer process and assessing the AMR dissemination pathways in environments impacted by human activity (Figure 3). The FCM’s significant advantages over the traditional plate-counting methods include speed, accuracy, and the possibility of quantification of viable non-culturable cells.

## 5. Perspectives for Further Contributions of FCM in Tackling AMR

One of the five overarching objectives of the WHO global action plan is to strengthen data on AMR through surveillance and research across human, animal, and natural environments, based on the premise that bacteria and genes can move without restriction between these sectors [96,97]. In this context, WHO developed under the GLASS umbrella, a simple surveillance cultivation-based methodology for assessing ESBL-*E. coli*, as an indicator for bacterial resistance to antibiotics, in samples collected from humans (hospital and community), food chains, and the environment. The FCM instrumental platforms could be included in such integrated trans-sectorial surveillance programs as they can increase the surveillance throughput and decrease the time to results. Currently, FCM-based methodologies have already demonstrated their ability to detect antibiotic resistance bacteria in human and animal biological samples [54,62,64,98], and in environmental samples [88,99,100]. Williams et al. (2017) [101] introduced an approach based on FCM for the detection and counting of *E. coli* O157:H7 pathogen directly in food (raw spinach) that proved more sensitive than the reference method. Additionally, total bacterial counts were determined directly in raw milk samples using an FCM method combined with SYTO64 and DiBAC4(3) [102]. These FCM developments could be extended to the rapid direct detection and enumeration of AMR bacteria in vegetables, meats, and processed foods. For this purpose, FCM protocols need to be further improved through simplification of the sample preparation coupled with cost-effective portable FCM devices that would facilitate its applications on animal farms.

Data integration from FCM analysis, protein analysis, and DNA sequencing could contribute to a comprehensive understanding of microbial communities’ responses to antibiotics. Li et al. (2022) [103] reported the usefulness of high throughput FCM and 16S rRNA gene sequencing in assessing the impact of low fosfomycin pressure over long periods of time on microbial communities from wastewaters, such information is of importance to both control spread and resistance evolution as these wastewaters are used on farmland [104]. Successful application of FCM cell counting in the evaluation of wastewater treatment processes has been reported [58,105,106]. FCM method in combination with nucleic acid stains revealed intact bacterial cells during photoelectrocatalysis treatment that could not be detected using cultivation [105]. Therefore, a more complete and exhaustive approach to assessing wastewater treatment efficiency is needed to avoid underestimating the risk of AR dissemination into the environment.

Microorganisms that co-habit in the same environment have relationships between themselves and with the host. The relationships between pathogens and co-habiting microorganisms can influence the response of pathogens to antibiotic treatment [107,108]. However, conventional AST methods do not take into consideration the presence of co-habiting species or how these might influence the acquisition of AMR. O’Brien et al. (2022) [109] reported that the polymicrobial nature of the cystic fibrosis airways impacts the clinical response to antibiotics, decreasing their efficiency against the targeted organism. FCM already proved to be a powerful tool for detecting the structural and functional alteration of complex aquatic microbial communities with high temporal resolution following antibiotic exposure [110]. Given the diverse polymicrobial communities associated with cystic fibrosis airway infections, FCM applications could be extended for an examination of how the antibiotic challenge affects the dynamics of polymicrobial consortium and, therefore, improve current antimicrobial treatment regimens in eradicating *P. aeruginosa* infections.

Total cell counting by FCM, without the need for cultivation, supports the development of rapid and accurate quantification of live and dead bacteria for disinfectant efficacy testing [111]. The advantage of FCM in counting microbial cells, regardless of their growth, aids the detection and quantification of non-cultivable cells. Thus, by detection of such bacterial populations on environmental surfaces, FCM could be used for investigation and control of bacteria with the potential to disseminate and produce nosocomial infections, permitting fast and effective actions.

AMR is acknowledged to be more enhanced in biofilms, in comparison with cells in a planktonic state of growth [108]. FCM could enable the investigation of how antibiotic treatment affects the population dynamics in this growth state. An experimental approach using FCM was developed by Mohiuddin et al. (2020) [112] for the quantification of persisters and viable but non-culturable (VBNC) cells, two phenotypic variants known to be highly tolerant to antibiotics, that can be expanded to different antibiotics and organisms. Such information will improve the antibiotic treatment of recurrent and chronic infections that are generally caused by biofilms formed by persister cells that are able to evade the host immune system and are highly resistant to antibiotic treatments [108].

One of the most outstanding contributions of FCM is the possibility to study the heterogeneity of bacterial populations and particular metabolic events following antibiotic challenges. Heteroresistance can result from mixed infection or clonal evolution within the same bacterial strain. It is a common phenomenon in *Mycobacterium tuberculosis* and it can rapidly lead to treatment failure and increased mortality [113]. Molecular drug susceptibility testing based on WGS alone can predict phenotypic drug resistance but with variable sensitivity. O’Donnell et al. (2019) [75] introduced a proof-of-concept protocol combining phenotypic FCM with WGS that enables rapid detection of drug heteroresistance in the clinical sputum of TB patients, which is essential for the prevention of drug resistance during early antibiotics treatment. The authors’ findings demonstrated the presence of mixed infection early in treatment, suggesting the need for comprehensive ASTs (phenotypic and/or genotypic) for the improvement of the antibiotic treatment regimens in multidrug-resistant tuberculosis, a global burden.

A major barrier to the advancement of FCM applications is the data acquisition and analysis and apparatus costs. Nevertheless, with technical progress, such as analytical software evolution, development of cost-effective portable imagining FCM devices, and knowledge about the potential of this technique, we are convinced that in the near future, the applications for this technique will increase. Göröcs et al. (2018) [99] reported a portable imaging FCM for cost-effective, high-throughput, and label-free analysis of natural water samples. Imaging FCM was used to detect and evaluate bacterial intracellular persistence [113]. The FCM technique has the advantage of analyzing the interaction of different bacterial isolates with live host cells, thus providing information about the tendency of some bacteria, associated with recurrent and chronic infections, to invade and persist within human host cells [114]. Moreover, machine learning algorithms could be used in combination with FCM analysis for rapid and accurate quantitative AST result prediction [51].

## 6. Conclusions

FCM is a powerful analytical tool with a wide range of potential applications in life sciences and biomedicine. The number of FCM applications in the field of microbiology has expanded rapidly, especially in response to AMR challenges. It has been demonstrated that FCM assays can make an important contribution to combating AMR. The development of novel AST frameworks that embed cytometry may inform clinicians in a timely manner. Additionally, the study of AMR ecology with FCM assays contributes to the understanding of AR acquisition and transmission mechanisms, sources, and routes of the spread of AMR. Future advances involving portable FCM instrumentation, coupled with simpler interpretation software, would support rapid and efficient AMR detection and quantification needed to implement AR monitoring programs and build AMR online databases. These data are critical for agencies and scientists to both manage AMR and assess public health risks.

## Figures and Tables

**Figure 1 microorganisms-11-01300-f001:**
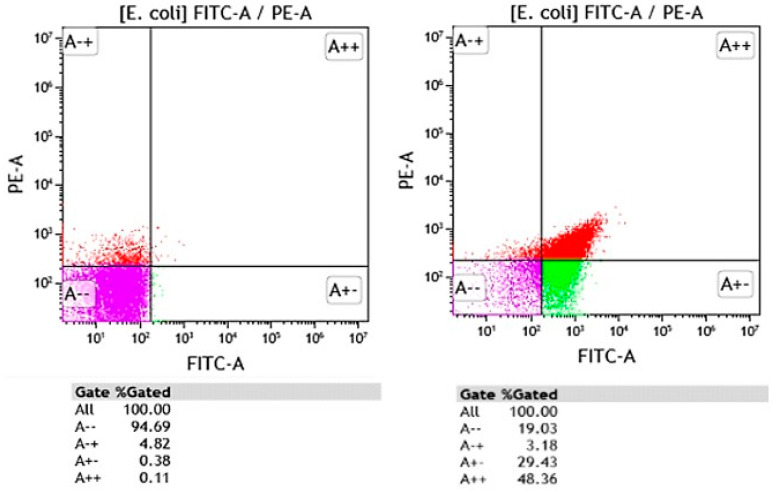
FCM graphs showing the *Escherichia coli* untreated (**left**) and *E. coli* exposed to a commercial biocidal product used in healthcare for surface disinfection (**right**). Bacteria were stained with propidium iodide (PI) and DiBAC4(3). Red fluorescence of PI was measured in PE-A channel whereas green fluorescence of DiBAC4(3) was detected in FITC-A channel. Discrimination between dead (A++ and A+), viable (A--), and injured cell (A+-) populations was performed on FITC-A vs. PE-A dot-plot plots. Purple region on FITC-A/PE-A dot-plots corresponds to viable populations with intact membrane integrity and membrane potential present, green region corresponds to damaged cell populations (loss of membrane potential and intact plasma membrane), the red region corresponds to dead cell populations (loss of membrane potential and irreversible damage to the plasma membrane).

**Figure 2 microorganisms-11-01300-f002:**
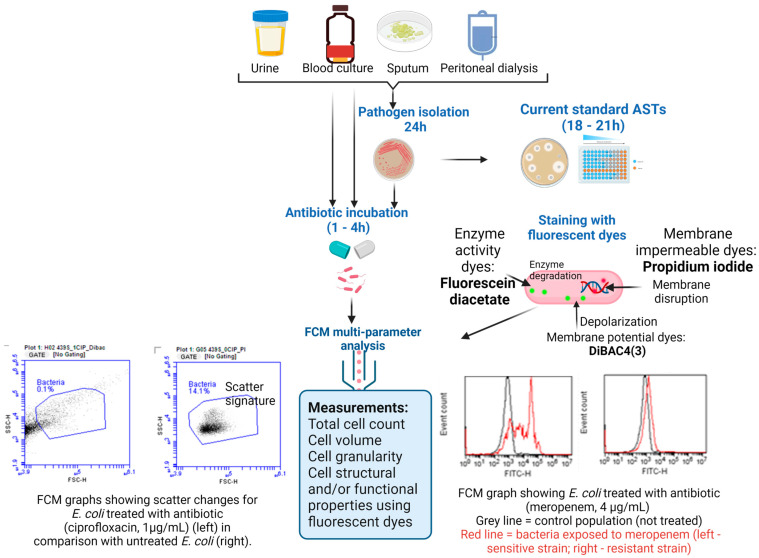
Schematic overview of the AST workflow using the FCM assay for rapid AMR diagnosis in clinical samples. (Figure created in Biorender).

**Figure 3 microorganisms-11-01300-f003:**
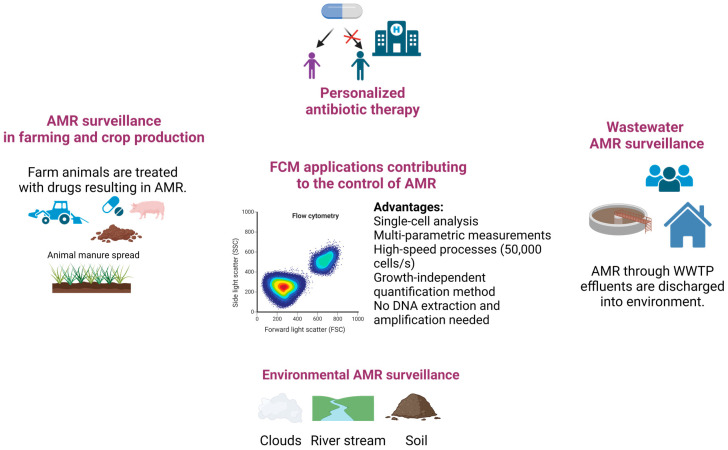
Schematic representations of FCM assays applied in controlling AMR in different sectors: humans, animals, and environment (Figure created in Biorender).

**Table 1 microorganisms-11-01300-t001:** Summary of FCM-based AST approaches developed in clinical microbiology.

Methods	Sample Analyzed (*n* = Number of Samples Tested)	Tested Bacteria (AMR Phenotype)	Principle of Detection	Tested Antibiotics	Time to Results	References
FCM coupled with MALDI TOF MS and VITEK 2	Urine samples (*n* = 211)	*Escherichia coli*	Cell-counting (cut-off 150 bacteria/mL)	Ampicillin, Amoxicillin/Clavulanic acid, Cefuroxime, Cefoxitin, Cefotaxime, Ceftazidime, Cefepime, Imipenem, Ertapenem, Gentamycin, Tobramycin, Nalidixic Acid, Ciprofloxacin, Fosfomycin, Nitrofurantoin, Cotrimoxazole	7 h	[54]
FCM	Urine samples (*n* = 107) *E. coli* strains (*n* = 19)	*E. coli*, *Klebsiella pneumoniae Proteus mirabilis*, *Pseudomonas aeruginosa*	Fluorescent dyes: DiBAC4(3)	Ceftriaxone, ciprofloxacin, nitrofurantoin, trimethoprim–sulfamethoxazole	4 h	[37]
FCM Fastinov	Positive blood cultures (*n* = 447)	Gram-positive and Gram-negative	Fluorescent dyes	Gram-negative: ampicillin, amoxicillin-clavulanic acid, cefotaxime, ceftazidime, ceftolozane-tazobactam, piperacillin–tazobactam, meropenem, imipenem, gentamicin, amikacin, ciprofloxacin, and colistin Gram-positive: ampicillin, penicillin, imipenem, vancomycin, linezolid, cefoxitin, and gentamicin	<2 h	[41]
FCM Fastinov	Spiked blood cultures (*n* = 204)	*Enterobacterales*, *Pseudomonas* spp., *Acinetobacter baumannii*	Fluorescent dyes	Colistin	<2 h	[39]
FCM Fastinov	Spiked blood cultures (*n* = 162)	*E. coli*, *K. pneumoniae*, *Enterobacter* ssp, *Serratia marcescens*, *Providencia* spp., *Morganella morgani*, *Proteus* spp.	Fluorescent membrane potential dye	Ceftolozane–tazobactam	<2 h	[40]
FCM	Blood spiked	*E. coli*, *K. pneumoniae*, *A. nosocomialis*	No		8 h	[38]
FCM and MALDI-TOF MS	Positive blood cultures (*n* = 238)	*Escherichia coli*, *Klebsiella pneumoniae*, *Pseudomonas aeruginosa*, *Enterobacter aerogenes*, *Acinetobacter baumannii*, *Klebsiella oxytoca*, *Proteus mirabilis*, *Enterobacter cloacae*, *Citrobacter freundii*, *Staphylococcus aureus*, *Staphylococcus saparophytics*, *Staphylococcus hominis*, *Enterococcus faecalis*, *Staphylococcus epidermidis*, *Staphylococcus simulans*, *Enterococcus faecium*, *Candida albicans*, *Candida tropicalis*, *Candida pseudotropicalis*, *Candida parapsilosis*	FDA PI	Ampicillin, vancomycin, cefotaxime, oxacillin, methicillin, ceftazidime amikacin, cefotaxime, ciprofloxacin	3 h	[55]
FCM	Blood culture samples	*E. coli*, *P. aeruginosa*, *S. aureus*	Antibiotic-induced changes in count rate	Ceftazidime, meropenem, tobramycin, oxacillin	5 h	[53]
Acoustic-enhanced FCM	Peritoneal dialysis effluent specimens	*Escherichia coli*, *Pseudomonas aeruginosa*, *Staphyloccocus aureus*, *Staphylococcus epidermidis*, *Klebsiella pneumoniae*	Live/DEAD™ Fixable Violet viability stain	Piperacillin–tazobactam, benzyl-penicillin, oxacillin, cefoxitin, vancomycin, teicoplanin, gentamicin, trimethoprim–sulfamethoxazole, daptomycin, erythromycin, clindamycin, amoxicillin, linezolid, ceftriaxone, ciprofloxacin, trimethoprim, cefepime, tigecycline, amikacin, aztreonam, amoxicillin-clavulanic acid, piperacillin–tazobactam, meropenem	4 h	[42]
MALDI-TOF and FCM	Clinical strains (*n* = 174)	*K. pneumoniae* (carbapenem resistant)	Fluorescent dyes: propidium iodide and thiazole orange	Meropenem	2 h	[56]
FCM	Clinical strains (*n* = 174)	*E. coli* and *K. pneumoniae*	YoPro-1	Colistin	3 h	[44]
Photoacoustic FCM	Clinical strains	*S. aureus*	Bacteriophage labeled with Direct Red 81	Daptomycin	4 h	[57]
FCM	Clinical and reference strains	*S. pneumoniae* *H. influenzae*	SYTO9 and PI	Penicillin G, cefotaxime	10 min	[45]
FCM	Reference strains (*n* = 6)	*Escherichia coli*, *Klebsiella pneumoniae*, *Pseudomonas aeruginosa*, *Staphylococcus aureus*, *Streptococcus pyogenes*, *Enterococcus faecalis*	Fluorescent dyes: acridine orange	Vancomycin, ciprofloxacin, levofloxacin, ceftriaxon, cefepime, amplicilin, piperacillin–tazobactam, trimethoprim–sulfamethoxazole, cefazolin, colistin, imipenem, gentamycin	4 h	[43]

**Table 2 microorganisms-11-01300-t002:** Comparison of current phenotypic and genotypic methods and the FCM method applied for AST.

Current Rapid ASTs	FCM Assays
Growth-dependent quantification methods Require isolation of the pathogens (18–24 h) ASTs on pure cultures (18–21 h) Detect only viable bacteria, culturable bacteria Quantify only live bacteria	Growth-independent quantification method Rapid ASTs directly in clinical samples (3–4 h) Rapid ASTs directly on pure cultures (1–3 h) Quantify both live bacteria and dead bacteria Detect viable culturable and non-culturable bacteria Detection of heteroresistance in polymicrobial infections
Genotypic AST methods Require DNA extraction and amplification Do not rely on microbial growth Detect only some specific genetic resistance markers	No DNA extraction required No amplification required

## Data Availability

Not applicable.

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
