# Peer review of "Current and Future Flow Cytometry Applications Contributing to Antimicrobial Resistance Control"

_microorganisms, 2023, doi:10.3390/microorganisms11051300_

Round 1

Reviewer 1 Report

A well-written, comprehensive, and complete review of FCM applications for the detection of pathogenic bacteria in clinical and environmental samples. The authors used tables and graphics to complement their statements which improve the clarity of the message. 

There are some minor points to address in this manuscript

line 129 - in FCM the scattering of the light is related to the relative size (FS) and complexity (SS). So, it is more correct to refer to the size rather than the volume of the cells.

Figure 1- I suggest improving the readability of the legend by adding the codification of the four quadrants in the text. (ex:  Discrimination between dead (A++ and A-+), viable (A--), and injured cell (A+-) populations.

Line 221-The citation number [53] is missing on "Filbrun et al (2022)"

Line 324- change "high throughout put FCM" to "high throughput FCM"

Author Response

Dear Reviewer 

I would like to thank Reviewer for taking the necessary time and effort to review the manuscript. I appreciate all your valuable comments and suggestions. Please find below my answers to your questions highlighted in red.

Point 1: line 129 - in FCM the scattering of the light is related to the relative size (FS) and complexity (SS). So, it is more correct to refer to the size rather than the volume of the cells.

Point 1: Thank you very much. I have corrected, I replaced „volume” with „size”.

Point 2: Figure 1- I suggest improving the readability of the legend by adding the codification of the four quadrants in the text. (ex:  Discrimination between dead (A++ and A-+), viable (A--), and injured cell (A+-) populations.

Point 2: Thank you vey much. I have added your suggestions in the Figure 1 legend

Point 3: Line 221-The citation number [53] is missing on "Filbrun et al (2022)"

Point 3: Thank you very much. I have added the citation [53] in the MS.

Point 4: Line 324- change "high throughout put FCM" to "high throughput FCM"

Point 4: Thank you very much. I made the modification suggested. Thanks.

Thank you very much for all your support!

Kind regards

Luminita

Reviewer 2 Report

The MS entitled "Current and future flow cytometry applications contributing to the antimicrobial resistance control" has an interesting purpose: to analyze FCM applications in microbiology to control AR.

The author aims to capture the reader's interest through a special design based on numerous recently published studies. Starting from the usual analyses in the microbiology lab followed by significantly performant ones, the author evidences the FCM high-speed execution (sections 3 and 4).

The Figures and Tables are relevant and well organized.

Some comments and suggestions are available as follows:

1. Lines 59-65 and 89-93: The reviewer suggests dividing the big phrases into 2 or 3 shorter ones to present the interesting data more clearly.

2. Lines 144-146 and 355-356:  Please check and reformulate the phrase for better understanding.

The same suggestion is available for lines 216-218.

3. Table 1 is not previously mentioned in the text of Section 4. Please, check and correct.

4. Section 4 has 2 extensive subsections: 4.1. and 4.2.  The reviewer suggests using different fonts in each subsection title, according to MDPI Suggestions for authors.

Moreover, the sub-subsections poorly marked on lines 172 and 194 could be considered titles and noted 4.1.1. and 4.1.2. according to MDPI instructions for authors. The same suggestion is available for sub-subsection 4.2. if the author agrees.

5. Figure 2 is complex and significant. 

However, the author is encouraged to increase the size of the letters for better visualization. 

6. Furthermore, the reviewer suggests a few data added in the MS text regarding FCM analysis of the other 2 specimens collected for analysis mentioned in Figure 2.  The same suggestion is available for "Personalized diagnosis therapy" in Figure 3.

7.  The reviewer observed the presence of a complex uncaptioned table with numerous data registered after the Conclusion section. The author is encouraged to write a Table caption and give a Table number. The Table should be suitably included in a  well-choice Section and before-mentioned in the MS text.

8. References:

Of 110 references, 76 have been published in the last 5 years (2018-2023)

However, the reviewer remarked on a combined editing style of references. The author is encouraged to attentively check and edit these data according to MDPI instructions for authors.

The same suggestion is available for the entire MS text to detect and correct all misprints.

Moderate English Language revision is required.

Author Response

Dear Reviewer,

I would like to take this opportunity to thank you for the effort and expertise that you contributed towards reviewing the article. Please find below the responses to your questions. Please find the revised MS attached, all the modifications are highlighted in red.

Point 1. Lines 59-65 and 89-93: The reviewer suggests dividing the big phrases into 2 or 3 shorter ones to present the interesting data more clearly.

Point 1: Thank you very much for your suggestion. I have organized the text in shorter sentences.

Point 2: Lines 144-146 and 355-356:  Please check and reformulate the phrase for better understanding.

The same suggestion is available for lines 216-218.

Point 2: Thank you very much. I have revised the phrases.

Point 3: Table 1 is not previously mentioned in the text of Section 4. Please, check and correct.

Point 3: Thank you very much. I have enclosed in the main text Table 2 (I have added another table before this section).

Point 4: Section 4 has 2 extensive subsections: 4.1. and 4.2.  The reviewer suggests using different fonts in each subsection title, according to MDPI Suggestions for authors.

Point 4: Thank you very much, I have added different fonts.

Moreover, the sub-subsections poorly marked on lines 172 and 194 could be considered titles and noted 4.1.1. and 4.1.2. according to MDPI instructions for authors. The same suggestion is available for sub-subsection 4.2. if the author agrees.

Point 4: Thank you very much, I agree with all your suggestions. I have added the titles 4.1.1. and 4.1.2. and mofidified the font for subsection 4.2.

Point 5: Figure 2 is complex and significant. 

However, the author is encouraged to increase the size of the letters for better visualization. 

Point 5: Thank you very much, I have increased the letter size.

Point 6:  Furthermore, the reviewer suggests a few data added in the MS text regarding FCM analysis of the other 2 specimens collected for analysis mentioned in Figure 2.  The same suggestion is available for "Personalized diagnosis therapy" in Figure 3.

Point 6: Thank you very much, I have added data on peritoneal fluid and sputum in the MS (4.1.3. Peritoneal dialysis and sputum). For personalized antibiotic therapy, I have mentioned the figure 3 in Section 4.1.

Point 7.  The reviewer observed the presence of a complex uncaptioned table with numerous data registered after the Conclusion section. The author is encouraged to write a Table caption and give a Table number. The Table should be suitably included in a  well-choice Section and before-mentioned in the MS text.

Point 7: Thank you very much, I have included the Table in Section 4.1.

Point 8. References:

Of 110 references, 76 have been published in the last 5 years (2018-2023)

However, the reviewer remarked on a combined editing style of references. The author is encouraged to attentively check and edit these data according to MDPI instructions for authors.

The same suggestion is available for the entire MS text to detect and correct all misprints.

Point 8: Thank you very much. The reviwer is correct, however not all the refferences older than 2018

Include data on FCM. The number of FCM articles older than 2018 are less than 10.

Thank you very much for your comments, I have check and edited the text.

Thank you very much for all your support!

Kind regards

Luminita
